# Forchlorfenuron-Induced Mitochondrial Respiration Inhibition and Metabolic Shifts in Endometrial Cancer

**DOI:** 10.3390/cancers16050976

**Published:** 2024-02-28

**Authors:** Kyukwang Kim, Negar Khazan, Rachael B. Rowswell-Turner, Rakesh K. Singh, Taylor Moore, Myla S. Strawderman, John P. Miller, Cameron W. A. Snyder, Ahmad Awada, Richard G. Moore

**Affiliations:** 1Division of Gynecologic Oncology, Department of Obstetrics and Gynecology, Wilmot Cancer Institute, University of Rochester Medical Center, Rochester, NY 14620, USA; negar_khazan@urmc.rochester.edu (N.K.); rachael_turner@urmc.rochester.edu (R.B.R.-T.); rakesh_singh@urmc.rochester.edu (R.K.S.); cameron_snyder@urmc.rochester.edu (C.W.A.S.); richard_moore@urmc.rochester.edu (R.G.M.); 2Department of Biology, University of Rochester, Rochester, NY 14627, USA; tmoore26@u.rochester.edu; 3Department of Biostatistics and Computational Biology, University of Rochester Medical Center, Rochester, NY 14642, USA; 4Department of Microbiology and Immunology, University of Rochester, Rochester, NY 14642, USA; johnp_miller@urmc.rochester.edu; 5Department of Gynecologic Oncology, Adventhealth, Orlando, FL 32804, USA; ahmad.awada.md@adventhealth.com

**Keywords:** forchlorfenuron, mitochondrial respiration, septins, AMPK, endometrial cancer

## Abstract

**Simple Summary:**

Our research focuses on the molecular mechanism of forchlorfenuron (FCF), a plant growth regulator widely used in agriculture to enhance fruit size and quality, and explored its potential in cancer treatment. Despite its known effects, the specific cellular and molecular mechanisms behind FCF’s actions have been unclear. Our study reveals a new mechanism: FCF inhibits mitochondrial respiration, affecting crucial cellular energy processes. This discovery sheds light on how FCF connects to the regulation of HIF-1α and glucose uptake observed in previous studies. Our findings not only deepen our understanding of FCF but also hint at the possibility of novel cancer therapies when combined with existing glycolytic inhibitors.

**Abstract:**

Forchlorfenuron (FCF) is a widely used plant cytokinin that enhances fruit quality and size in agriculture. It also serves as a crucial pharmacological tool for the inhibition of septins. However, the precise target of FCF has not yet been fully determined. This study reveals a novel target of FCF and elucidates its downstream signaling events. FCF significantly impairs mitochondrial respiration and mediates metabolic shift toward glycolysis, thus making cells more vulnerable to glycolysis inhibition. Interestingly, FCF’s impact on mitochondrial function persists, even in cells lacking septins. Furthermore, the impaired mitochondrial function leads to the degradation of HIF-1α, facilitated by increased cellular oxygen. FCF also induces AMPK activation, suppresses Erk1/2 phosphorylation, and reduces the expression of HER2, β-catenin, and PD-L1. Endometrial cancer is characterized by metabolic disorders such as diabetes and aberrant HER2/Ras-Erk1/2/β-catenin signaling. Thus, FCF may hold promise as a potential therapeutic in endometrial cancer.

## 1. Introduction

Cytokinins are a class of growth hormones that promote cell division and differentiation in plants. They can be classified into two distinct groups: phenylurea-based and purine-based cytokinins. The initial breakthrough in the phenylurea class came in 1955 when Shantz and Steward identified the cytokinin activity of 1,3-diphenylurea within coconut milk extract [1]. Building upon this, in 1966, Bruce et al. synthesized and assessed over 500 urea and thiourea derivatives, uncovering the cytokinin activity of *N*-phenyl-*N′*-(4-pyridyl)urea [2]. This led to subsequent structural modifications and the discovery of the highly active cytokinin compound *N*-phenyl-*N′*-(2-chloro-4-pyridyl)urea, also known as forchlorfenuron (FCF), KT-30, or CPPU [3]. FCF is widely used in agriculture and horticulture to increase the size and quality of fruits, like kiwifruit, watermelon, and grapes. It works in synergy with natural auxins to promote cell division and lateral growth. Notably, FCF’s impact extends beyond plants; in 2004, Iwase et al. revealed that FCF disturbs cytokinesis in yeast and causes the ectopic structure of septins, a family of GTP-binding cytoskeletal proteins [4]. Since then, FCF has served as a major pharmacological tool to perturb septin function in various models, from yeast to mammalian cells. Despite its primary role in septin perturbation, some research hints at alternative (non-septin) targets of FCF [5,6,7].

In mammalian cells, treatment with FCF has been observed to exert a significant influence on various biological activities, including glucose uptake [8], Ca^2+^ influx [9], exocytosis [10], and mitochondrial fragmentation [6]. The evaluation of FCF in cancer has just begun yet has already been found to promote the degradation of the proto-oncogenes HIF-1α [11] and ErbB2/HER2 [12] and to reduce HE4 expression [13]. FCF also blocks hallmark features of cancer, such as proliferation, anchorage-independent growth, invasion, and migration [11,14], thus preventing tumor growth in animals [15]. However, the mechanisms that underpin the cellular and molecular effects of FCF remain unclear. In this study, we demonstrate that FCF targets mitochondria and alters cellular energetics and that FCF-induced HIF-1α regulation and glucose uptake, as seen in previous studies [8,11], are associated with impaired mitochondrial respiration due to FCF treatment. We also highlight the therapeutic potential of FCF in the treatment of endometrial cancer.

## 2. Materials and Methods

### 2.1. Cell Culture and Materials

HEC1A and RL95-2 cells were purchased from the American Type Culture Collection. MFE280 and MFE296 cells were purchased from the German Collection of Microorganisms and Cell Cultures. They were authenticated by the vendor and passaged in our lab for less than 6 months. HEC1A (McCoy’s 5A), RL95-2 (DMEM/F-12), MFE280 (MEM), and MFE296 (MEM) cells were maintained in the respective medium in brackets supplemented with 10% FBS and placed in a humidified incubator at 37 °C with 5% CO_2_. 143B and 143Bρ^0^ human osteosarcoma cells were a gift of Dr. Giovanni Manfredi (Cornell University). All other cells were maintained as previously described [13,16]. Chemicals were purchased from vendors as follows: FCF (Abcam. 152 Grove Street, Waltham, MA, USA), lapatinib (LC Laboratories. 165 New Boston St, Woburn, MA, USA), CI-1040 (Selleckchem. 300 Industry Drive, Pittsburgh, PA, USA), 2-deoxyglucose (TCI. 9211 North Harborgate Street, Portland, OR USA), pimonidazole (Tocris. 614 McKinley Place NE, Minneapolis, MN, USA), AICAR (Selleckchem), rotenone (Cayman. 1180 East Ellsworth Road, Ann Arbor, Michigan, USA), rotenone as in a mixture with antimycin A (Agilent. 5301 Stevens Creek Blvd, Santa Clara, CA, USA), thapsigargin (Cayman), A769662 (LC Laboratories), oligomycin (Sigma Aldrich. PO Box 14508, Saint Louis, MO, USA), BAY87-2243 (Selleckchem). Further, 1000× FCF stock solution was made in dimethyl sulfoxide (DMSO). The final concentration of DMSO in culture medium did not exceed more than 0.1%.

### 2.2. Antibodies

Antibodies were obtained from venders as follows: HIF-1α (Novus biologicals. 10771 E Easter Ave, Centennial, CO, USA. NB100-479), Raptor (Proteintech. 5500 Pearl St # 400, Rosemont, IL, USA. 20984-1-AP), anti-pimonidazole (Hypoxyprobe. 121 Middlesex Turnpike, Burlington, MA, USA. 4.3.11.3 mouse MAb), septin-2 (Novus biologicals, NBP2-75660), septin-5 (Proteintech, 11631-1-AP), septin-11 (Proteintech, 14672-1-AP). Other septin antibodies were purchased from Sigma Aldrich (septin-6, HPA005665; septin-7, HPA029524; septin-9, HPA042564; septin-10, HPA056304). All other antibodies were purchased from Cell Signaling Technology, (3 Trask Lane, Danvers, MA, USA): AMPKα (5831), p-AMPKα (2535), p-Raptor (2083), ACC (3676), p-ACC (11818), α-tubulin (2144), acetyl-α-tubulin (5335), β-catenin (8480), HER2 (4290), p-HER2 (2247), Erk1/2 (9102), p-Erk1/2 (4370), cleaved PARP (5625), cleaved caspase 7 (9491), and PD-L1 (13684).

### 2.3. Cell Proliferation, Population, and Cytotoxicity Assay

Cell population and proliferation were determined by the Sulforhodamine B (SRB) assay [17], BrdU Cell Proliferation (Cell Signaling Technology, 6813), or MTS Assay (Promega. 2800 Woods Hollow Road, Madison, WI, USA. G3580) according to manufacturer’s recommendations. For the measurement of cell cytotoxicity (Figure 2D), we used the CellTox Green Cytotoxicity Assay (Promega, G8741) and followed the manufacturer’s protocol.

### 2.4. Measurement of Glucose Uptake and Lactate

To assess cellular glucose uptake by FCF or mitochondrial inhibitors, we used Glucose Uptake-Glo Assay (Promega, J1341) and followed the manufacturer’s protocol. The assay is based on the bioluminescent detection of 2-deoxyglucose-6-phosphate (2-DG-6-P). Once inside cells, 2-DG, a D-glucose mimic, forms 2-DG-6-P, which is not further metabolized to fructose-6-P, so it accumulates in cells. Briefly, following treatment with FCF or mitochondrial inhibitors, MFE296 cells were incubated with 1 mM 2-DG in 10% FBS in PBS for 10 min at room temperature. The reaction was then stopped and neutralized. 2-DG-6-P detection solution was prepared, added to cells, and incubated for 20 min at room temperature before 2-DG-6-P was measured by a luminometer. To measure lactate levels in culture medium, we used L-Lactate Assay Kit (Cayman, 700510) and followed the manufacturer’s protocol.

### 2.5. Measurement of Adenine Nucleotides

Intracellular adenine nucleotides in MFE296 cells were extracted utilizing the boiling water method [18]. AMP, ADP, and ATP concentrations were measured using ATP/ADP/AMP Assay (Biomedical Research Service, A-125) following the manufacturer’s recommendations. The assay is based on the ATP-dependent luciferase reaction for bioluminescence measurement. Cellular AMP and ADPs were converted into ATPs by adenylate kinase and pyruvate kinase, respectively. The levels of each nucleotide (AMP, ADP, and ATP) were estimated per the manufacturer’s protocol.

### 2.6. Measurement of Intracellular Ca^2+^ and ROS

For the measurement of Ca^2+^, MFE296 cells were seeded in 96-black-well plates and loaded with a solution containing Fluo-8 dye and Pluronic F147 Plus in 1X Hank’s buffer with 20 mM HEPES. The plate was incubated according to the manufacturer’s protocol (Fluo-8 No Wash Calcium Assay, Abcam, ab112129). Once the vehicle, FCF, or thapsigargin were added to appropriate wells, fluorescence intensity was monitored at λ_ex_/λ_em_ = 490/525 nm. For the detection of reactive oxygen species (ROS), MFE296 cells were seeded in 96-black-well plates and loaded with 25 μM of carboxy-H_2_DCFDA (Invitrogen. 5791 Van Allen Way, Carlsbad, CA, USA. C400). After 30 min, cells were exposed to the indicated conditions and incubated for 2 h. Fluorescence intensity was monitored according to the manufacturer’s protocol.

### 2.7. Mitochondrial Respiration and Respiratory Chain Complex Activities

Oxygen consumption rate (OCR) and extracellular acidification rate (ECAR) were measured using the Seahorse XFe96 Analyzer. Cells were seeded in Seahorse cell culture plates and allowed to adhere. After overnight incubation, culture medium was switched to Seahorse XF DMEM medium (Agilent, 103575) and supplemented with 10 mM glucose, 1 mM pyruvate, and 2 mM glutamine, followed by real-time monitoring of OCR and ECAR. To measure the activity of mitochondrial complex I or complex V of isolated mitochondria, we used MitoCheck Complex I or Complex V Activity Assay kit (Cayman, 700930 and 701000).

### 2.8. Transfection

MFE296 cells were transiently transfected with septin-2, septin-5, septin-7, or non-targeting siRNAs (Santa Cruz Biotechnology. 10410 Finnell St, Dallas, TX, USA. sc-40936, sc-36478, sc-36480, and sc-37007) using Lipofectamine 3000 (Thermo Fisher Scientific. 168 Third Avenue, Waltham, MA, USA. L3000015) per the manufacturer’s protocol. To establish a stable knockout of septin-7, MFE296 cells were transfected with septin-7 Double Nickase plasmid (Santa Cruz Biotechnology, sc-401655) and were selected under puromycin pressure (1.0 μg/mL) for 1–2 weeks. Stable knockout of AMPKα_1_ and AMPKα_2_ (PRKAA1 and PRKAA2) was conducted using the CRISPR/Cas9 system. Constructs encoding Cas9 and gRNAs targeting PRKAA1 or PRKAA2 were a gift from Reuben Shaw [19] (Addgene: 74374, 74375, 74376, and 74377). MFE296 cells were transfected with both gRNA pairs. Following puromycin selection, single-cell cloning was carried out by serial dilution in 96-well plates. Individual clones were harvested and screened by Western blotting. Full Western blot images can be found at File S1. Clones lacking both AMPKα_1_ and AMPKα_2_ were selected for this study.

### 2.9. Immunoblotting, Enzyme-Linked Immunosorbent Assay, and Quantitative RT-PCR

Following indicated treatments, cell extracts were prepared using Cell Lysis Buffer (Cell Signaling Technology, Cat# 9803) and subjected to gel electrophoresis in a NuPAGE 4–12% Tris-Bis gel (Invitrogen, NP0322) with MES SDS running buffer or in a NuPAGE 3–8% Tris-Acetate gel (Invitrogen, EA03752) with Tris-Acetate SDS running buffer. Resolved proteins were wet transferred onto a nitrocellulose membrane (Bio-Rad. 1000 Alfred Nobel Dr, Hercules, CA, USA. 1620097) by Mini Blot Module (Invitrogen, B1000). Membranes were blocked by 5% nonfat milk in TBS-Tween 20 before incubation with primary and secondary antibodies (separately listed). Where applicable, membranes were stripped by OneMinute Western Blot Stripping Buffer (GM Biosciences. 5350 Partners Ct. Suite C, Frederick, MD, USA. GM6001) before reprobing. Quantitative RT-PCR was conducted as previously described [13] using TaqMan assays (Applied Biosystems. 5791 Van Allen Way, Carlsbad, CA, USA.) as follows: HIF1A (Hs00153153), PDK1 (Hs01561850), SLC2A1 (Hs00892681), SLC2A4 (Hs00168966), and TBP (reference gene, Hs00427620).

### 2.10. Animal Study

AN3CA cells (600,000 cells/animals) were implanted subcutaneously in NSG mice (N = 10) in a suspension of cold DMEM medium and Matrigel (Corning) mixture (1:1). NSG mice were obtained from in-house breeding colonies. When tumors became palpable in all ten mice, animals were divided into two groups: one group received vehicle, which was a 100 µL solution containing PBS and DMSO in a 1:1 ratio, while the other group received FCF treatment (25 mg, IP, Monday–Friday) using a 100 µL solution made up of 50 µL PBS and 50 µL DMSO. Mice were maintained in University of Rochester animal facility. Tumor volumes were measured using digital calipers at treatment initiation and days 2, 3, 4, and 7 using the formulation (length × width^2^)/2. The average change in tumor volume from the baseline volume at treatment initiation was compared between treatment groups using a mixed effects model. Treatment, day, and their interaction were included as fixed effects. A random effect for animals using unstructured covariance accounted for the correlated observations on each animal over time. The interaction effect F-test is the assessment of differential tumor growth for FCF vs. vehicle.

### 2.11. Genomics Data, Statistical Analysis, and Abstract Graphic

Genomics data (The Cancer Genome Atlas dataset, number of samples = 541) were obtained from the Human Protein Atlas [20], available from http://www.proteinatlas.org, accessed on 1 January 2024. All the statistical analyses were performed using GraphPad Prism. For the survival analysis, the Mantel–Cox test was used to calculate *p*-values. For all the others, we performed a two-tailed unpaired *t*-test. Abstract graphic was created with ChemDraw (PerkinElmer, Waltham, MA, USA) and BioRender.com, accessed on 1 January 2024.

## 3. Results

### 3.1. FCF Affects the Proliferation of Endometrial Cancer Cell Lines

Septins are putative targets of FCF. A range of 50–100 µM of FCF, as employed in the current study, has been demonstrated to trigger responses in mammalian septins [10,11,21]. As such, we explored the relationship between septins and patient mortality in endometrial cancer. The Cancer Genome Atlas (TCGA) database analysis revealed that the expression of septin-2 and septins 6–11 is notably elevated in comparison to other septins (Figure 1A). These high expressors were further analyzed using Kaplan–Meier survival analysis and evaluated for their associations with survival rates. Septin-2, -8, -10, and -11 show a trend of poor prognosis (*p* < 0.05) (Figure 1B), while the relationship with other septins is not significant.

We next evaluated the effects of FCF on the proliferation of endometrial cancer cell lines. The population of MFE296 cells without FCF increased by 91% (24 h) in the SRB assay, while FCF at 100 μM inhibited their proliferation by 59% (Figure 2A). At higher doses (≥150 μM), cell proliferation was completely blocked. This observation was similar in the BrdU assay (Figure 2B). The anti-proliferative effects of FCF were also observed in other endometrial cancer cell lines (KLE, AN3CA, HEC1A, RL95-2, and MFE280). In the absence of FCF, each cell line grew rapidly, but when exposed to FCF (100 μM, 48 h), the proliferation of most cell lines was brought to a standstill (Figure 2C). Our data show that FCF is not a potent inducer of apoptosis (Figure 2F,G), nor of cytotoxicity (Figure 2D), in MFE296 cells. Notably, the MTS assay, relying on the measurement of mitochondrial metabolic rates [22], failed to detect the inhibitory effects of FCF. In the MTS assay, the optical density of FCF-treated cells was comparable to that of untreated cells (Figure 2E). Direct cell counting and the SRB assay were still able to recapitulate FCF’s marked reduction in cell proliferation.

### 3.2. FCF Affects Oncogenic Pathways in Endometrial Cancer

Next, we investigated the therapeutic potential of FCF in endometrial cancer, focusing on its impact on HER2, Ras, and β-catenin, which are frequently activated in this type of cancer [23]. Erk1/2 was included as a downstream component of the HER2-Ras-Raf-MEK-Erk1/2 signaling cascade. Upon treating KLE cells with FCF, we observed a reduction in β-catenin expression (Figure 3A). Concurrently, FCF led to an increase in tubulin acetylation (Appendix A), which may be linked to septin disruption, as described in a previous study [24]. Previous studies showed that FCF induced HER2 downregulation [12,25]. Our data also confirmed that FCF induced HER2 downregulation in AN3CA and KLE cells (Figure 3A). Interestingly, this phenomenon did not occur in MFE296 cells, while the activation of HER2 was still inhibited by FCF (Figure 3B). Therefore, we explored how HER2 and Erk1/2 impact cell growth during FCF treatment in MFE296 cells. The inhibition of Erk1/2 by the MEK inhibitor CI-1040 resulted in a notable suppression of cell growth in MFE296 cells (Figure 3C). Interestingly, despite HER2-inhibitor lapatinib treatment, MFE296 cells displayed resistance, as it failed to inhibit Erk1/2 phosphorylation (Figure 3D) and did not reduce cell growth (Figure 3C). In contrast, lapatinib was effective in KLE cells in both aspects. These findings suggest that FCF may influence Erk1/2 in MFE296 cells via a novel pathway independent of HER2.

Due to the high frequency of microsatellite instability and elevated mutational burden, subtypes of endometrial cancers are immunogenic, and immune checkpoint blockade against the PD-1/PD-L1 axis has been explored as a viable therapeutic option [23,26]. First, we looked for PD-L1-positive cell lines. Among seven different endometrial cancer cell lines tested (AN3CA, ECC-1, HEC1A, KLE, RL95-2, MFE280, and MFE296), strong PD-L1 expression was found in AN3CA cells, whereas all the others showed weak to no expression. Because the MEK-Erk1/2 signaling pathway may contribute to PD-L1 expression [27], we evaluated the effects of MEK inhibition on PD-L1. We found that CI-1040 does induce a dose-dependent reduction in PD-L1 expression (Figure 3E). Similarly, FCF induced downregulation in both PD-L1 and Erk1/2 phosphorylation in AN3CA cells.

### 3.3. FCF Alters Energy Metabolism in Endometrial Cancer Cells

When KLE cells were exposed to FCF overnight, a noticeable color change occurred in the culture medium (Figure 4A). This change strongly suggests medium acidification and increased glycolysis. To further explore this drug’s impact, we conducted an analysis of lactate levels in the medium, a known byproduct of glycolysis. The results confirmed that FCF enhanced lactate secretion in KLE (Figure 4B) and MFE296 cells (Appendix A). Concurrently, FCF also increased glucose uptake in MFE296 cells (Figure 4C). This shift towards glycolysis might be a response to an energy crisis, such as cellular ATP depletion. To investigate further, we examined the impact of FCF on ATP levels. After incubating MFE296 cells with FCF, we observed a significant decrease in ATP levels by 60.1% (Figure 4D). Furthermore, FCF led to an increase in the AMP to ATP ratio in MFE296 (Figure 4E) and KLE cells (Appendix A). To inquire if FCF-treated cells rely more on glycolysis for their survival, the cells were co-treated with 2-deoxyglucose (2-DG). 2-DG competitively blocks glucose transport and, once entered, is not further metabolized, thus leading to the inhibition of glycolysis. The cells co-incubated with FCF and 2-DG produced higher levels of cleaved PARP, a marker of apoptosis (Figure 4F), compared to that of FCF or 2-DG as a single agent. This drug combination also had a greater effect in inhibiting cell proliferation (Figure 4G). In addition, similar results were observed when cells were exposed to a glucose-deprived cell culture medium instead of 2-DG treatment (Figure 4F,H). Together, our data suggest that FCF exposure alters the energy metabolism of cancer cells.

### 3.4. FCF Activates the AMPK Signaling Pathway Independent of Oxidative Stress or Ca^2+^ Levels

AMPK, a highly conserved and crucial energy sensor, is responsible for maintaining energy homeostasis within cells. Its activation is triggered by an increase in the ratio of AMP/ATP and/or ADP/ATP. Considering that FCF reduces cellular ATP levels and elevates the AMP/ATP ratio, we aimed to explore the impact of FCF on the AMPK pathway. Our study revealed that in MFE296 cells, FCF induced AMPK phosphorylation at Thr172, which plays a crucial role in AMPK activity (Figure 5A). FCF treatment also led to the activation of ACC and Raptor, which are downstream targets of AMPK. FCF induced AMPK phosphorylation in a dose-dependent manner (Figure 5B). Consistently, FCF was found to activate AMPK in KLE cells as well (Appendix A). We conducted a preliminary in vivo efficacy study of FCF in a human endometrial cancer cell xenograft mouse model (Figure 5C). FCF treatment inhibited the progression of tumor growth on average based on a statistically significant interaction between treatment and day with respect to the change from volume at treatment initiation (F-test *p*-value = 0.02). On day 7, the average change in tumor volume was 874 (SE = 330) mm^3^ less in the FCF-treated group. Furthermore, there was evidence of a higher frequency of AMPK activation in tumors in the FCF treatment group. Next, we explored if FCF affects alternative pathways that do not rely on nucleotide ratios for AMPK activity. Because the levels of intracellular Ca^2+^ could impact AMPK activity [28], we loaded MFE296 cells with Fluo-8, a fluorescent Ca^2+^-sensitive probe, and monitored the levels of Ca^2+^ following FCF injection. We included thapsigargin, a Ca^2+^-ATPase inhibitor, as a positive control. Thapsigargin rapidly increased Ca^2+^ levels and peaked at ~120 s, after which it began to decrease until 300 s (Figure 5D). Up to the time point we monitored, FCF did not increase Ca^2+^ levels.

Although the metabolic and physiological conditions associated with nucleotide ratios or Ca^2+^ levels are the primary factors contributing to AMPK activity, some studies have reported the possible role of reactive oxygen species (ROS) in regulating AMPK [29,30]. Thus, we evaluated if FCF treatment produces ROS in cells. To measure cellular ROS, MFE296 cells were loaded with carboxy-H_2_DCFDA, a cell-permeable and non-fluorescent compound. Upon oxidation by ROS, the compound produces a fluorescent DCF dye inside cells. We found that incubating MFE296 cells with FCF did not increase ROS, while H_2_O_2_ rapidly increased DCF fluorescence in MFE296 cells (Figure 5E). Trolox is a vitamin E analog with a potent antioxidant capacity. Trolox, at the dose that effectively quenches H_2_O_2_-generated ROS, did not suppress FCF-induced AMPK phosphorylation (Figure 5F).

### 3.5. FCF Affects Mitochondrial Respiration

Mitochondria serve as the primary source of energy production within cells. Inhibiting mitochondrial energetics may trigger the activation of AMPK signaling. Consequently, we delved into the bioenergetic properties of FCF-treated cells using the Seahorse analyzer. This instrument allowed us to measure the oxygen consumption rate (OCR) and extracellular acidification rate (ECAR), which represent the rate of mitochondrial respiration and cellular glycolysis, respectively. We found that FCF decreases OCR and increases ECAR in KLE, MFE296, and other cancer cell lines (Figure 5G and Appendix A), suggesting reduced mitochondrial respiration and a metabolic shift toward glycolysis. Complexes I and V of the mitochondrial respiratory chain are common targets of several AMPK activators [31]. We observed that FCF at 100 μM moderately suppressed complex I activity (Figure 5H), while a higher dose (300 μM) showed much stronger inhibition. Furthermore, FCF potently inhibited complex V activity, even at 100 μM. Next, we explored the effects of FCF without mitochondrial respiration using 143Bρ^0^ cells, which were generated through chronic exposure of 143B to ethidium bromide, resulting in the depletion of mitochondrial DNA (Appendix A) and consequent respiratory impairment (Figure 5J). FCF inhibited OCR and reduced cell proliferation in parent 143B cells, and its combination with glucose depletion exhibited synergistic effects (Figure 5I,J). However, 143Bρ^0^ cells displayed only baseline OCR, unaffected by FCF. Interestingly, the proliferation of 143Bρ^0^ remained vulnerable to FCF, suggesting alternative, non-respiratory targets for FCF also exist (Figure 5I).

Previously, FCF was found to inhibit HIF-1α in a PC-3 prostate cancer cell line [11]. HIF-1 target genes were also suppressed by FCF. Likewise, we found that FCF affects HIF-1α signaling in MFE296 cells; low-oxygen conditions (5% O_2_) raised HIF-1α expression, while FCF notably suppressed it (Figure 6A), along with downstream targets such as PDK1 and GLUT1 (Figure 6B). Since FCF inhibits mitochondrial respiration, we tested other inhibitors such as BAY87-2243 (Complex I) and oligomycin (Complex V) on HIF-1α. Western blot analysis showed that they also inhibit HIF-1α, akin to FCF (Figure 6C). It has been suggested that by inhibiting mitochondrial respiration, more oxygen could become available for the degradation of HIF-1α [32,33]. In order to test this hypothesis, we employed pimonidazole. This compound becomes active in response to oxygen deficiency and forms immunogenic adducts, which can be identified using a specialized monoclonal antibody [34]. MFE296 cells loaded with pimonidazole and exposed to 5% O_2_ clearly yielded pimonidazole adducts (Figure 6D). These adducts disappeared when cells were co-treated with FCF or other mitochondrial inhibitors. This suggests that FCF triggers HIF-1α degradation by increasing cellular oxygen through impaired mitochondrial respiration. Additionally, mitochondrial inhibition increased glucose uptake. Oligomycin and BAY87-2243, along with FCF, elevated glucose uptake in MFE296 cells (Figure 6E).

Next, we explored the impact of mitochondrial inhibition on Erk1/2 signaling. We found that FCF, along with other mitochondrial inhibitors, effectively decreased Erk1/2 phosphorylation in MFE296 cells (Figure 6F). We then asked if AMPK is required for FCF to regulate Erk1/2 signaling [35]. To do so, we generated double-knockout (DKO) clones that lack both the AMPKα1 and AMPKα2 isoforms within MFE296 cells. Treating wild-type (WT) cells with FCF led to clear activation of AMPK and its substrates (ACC and Raptor); this activation vanished in one DKO clone and reduced in the other. Interestingly, the second DKO clone still exhibited a faint residual phosphorylation of ACC and AMPK for unclear reasons (Figure 6G). We noticed that even in AMPK knockouts, FCF treatment continued to suppress Erk1/2 phosphorylation, implying a regulatory effect of FCF on Erk1/2 that is independent of AMPK (Figure 6G). In support of this, we found that A769662, an allosteric AMPK activator, failed to modulate Erk1/2 (Figure 6H). Interestingly, AICAR, another AMPK activator that mimics cellular AMPs, consistently inhibited Erk1/2 phosphorylation in both AMPK wild-type and knockout cells.

Next, considering the essential role of septin-2 and -7 in Erk1/2 regulation in MDA-MB-231 cells [14], we investigated the effects of septin-2 and -7 inhibition on Erk1/2 activity in MFE296 cells. Transient knockdown of septin-2 and -7 using siRNAs, indeed, downregulated Erk1/2 phosphorylation (Figure 7A top). Additionally, septin-7 knockdown caused the downregulation of other septins (Figure 7A bottom); this phenomenon of septin coregulation has been observed previously [36]. Interestingly, a recent study demonstrated that knocking out septin-7 does not inactivate Erk1/2 in HT-29 cells [7], which adds complexity to how to interpret septin-mediated Erk1/2 regulation. Indeed, we came across a similar issue when working with a septin-7 knockout that we generated in MFE296 cells. Following antibiotic-mediated cell selection, septin-7-depleted cells did not exhibit Erk1/2 inactivation (Figure 7B top).

### 3.6. FCF Causes Septin-Independent Inhibition of Mitochondrial Respiration

Lastly, we explored the role of septins in FCF-mediated mitochondrial respiratory inhibition. To address this, we modified septin-7 knockout cells. Because these cells still expressed significant levels of septin-2 and -5 (Figure 7C, left), we employed specific siRNAs for further suppression, achieving more comprehensive downregulation over septin-2, -5, -6, -7, -10, and -11. Notably, when these cells were treated with FCF, it was observed that FCF retained its capacity to diminish OCR while increasing ECAR (Figure 7C, right). Similar results were observed in MFE296 cells with simultaneous septin-2, -5, and -7 depletion. These findings collectively suggest that the FCF-induced inhibition of mitochondrial respiration may take place irrespective of septins.

## 4. Discussion

Metabolic disorders such as diabetes are known to elevate the risk of endometrial cancer [37]. Several studies have indicated that the anti-diabetic drug metformin can enhance survival in patients with this disease [38]. Interestingly, the mechanism of action for metformin involves the inhibition of mitochondrial respiration and activation of AMPK [39]. Considering the possible association between septins and survival rates in this cancer type, combining the mitochondrial inhibitory function of FCF with its ability to disrupt septins may offer additional benefits in managing endometrial cancer.

In this study, we found that FCF, along with other mitochondrial respiratory chain inhibitors, increases critical oxygen in cells, thus leading to the degradation of HIF-1α during hypoxic stress (Figure 6D and Figure 8). This aligns with other studies that elucidate the significance of mitochondrial inhibition in the regulation of HIF-1α. [32,40,41]. FCF also triggered a metabolic shift toward glycolysis (Figure 4F–H, Figure 6E and Figure 8), possibly acquiring compensatory cytoprotection during this energy crisis [42,43]. Our observations on HIF-1α and glucose uptake appear to be a direct consequence of impaired mitochondrial function due to FCF. In comparison, loss of HIF-1α generally reduces glucose uptake in the canonical mechanism [44]. Our findings may further inform how FCF enhances glucose uptake in fibroblasts [8]. This study suggests that septin-7 inhibits the fusion process between GLUT4 storage vesicles and the plasma membrane. Consequently, when septin-7 is knocked down, glucose uptake increases. The cited paper employed FCF to simulate septin disruption, but its specific impact on vesicle trafficking remains unclear. Moreover, it is important not to dismiss the possibility of FCF also affecting mitochondrial respiration as a potential contributing factor [11].

Erk1/2 is a central protein that conveys mitogenic and growth factor signals to cells, largely through the receptor tyrosine kinase (RTK)-Ras-Raf-MEK signaling cascade. The activation of Erk1/2 is known to play a crucial role in cell growth, survival, proliferation, migration, and differentiation [45]. Our study demonstrates that FCF can inhibit both HER2 expression and its phosphorylation, effectively targeting two critical regulatory mechanisms. Consequently, FCF can contribute to a reduction in Erk1/2 activity. However, it should be noted that the potential effects of other RTKs, which are known to associate with septins, have not been fully assessed in this study and could also be influencing Erk1/2 protein [46,47].

The following question still remains: does FCF’s influence on cells correlate with its capacity to induce septin disruption and subsequent downstream signaling cascades? To date, FCF is the only commonly available pharmacological septin inhibitor. FCF has been shown to cause ectopic septin structures [5] and to alter septin assembly and dynamics in mammalian cells [21], replicating the effects of septin depletion [12,14]. However, compelling evidence exists for the non-septin effects of FCF [5,6,7]. Our findings also indicate that mitochondrial respiratory inhibition by FCF might not be dependent on septins (Figure 7C). Furthermore, the impact of septins on Erk1/2 signaling remains uncertain. While transient septin-7 knockdown resulted in Erk1/2 inactivation, this effect was not observed in stable septin-7 knockout (Figure 7A,B). Rather, FCF-induced mitochondrial respiratory inhibition (Figure 6F) and an increase in AMPs (i.e., AICAR in Figure 6H) are possibly playing a role in Erk1/2 inactivation in MFE296 cells. Up until now, the availability of a highly selective pharmacologic inhibitor or functional assay for investigating septins was quite limited. Therefore, when evaluating septins, it is essential to approach data obtained from FCF or other genetic approaches (e.g., siRNA) with caution and meticulous interpretation. Nevertheless, our study does not attempt to discount the continuing importance of FCF but, rather, promote its therapeutic potential, particularly in endometrial cancer. A more selective, potent, and specific pharmacological inhibitor against septins is needed to produce reliable and interpretable data.

## 5. Conclusions

In conclusion, our study reveals that FCF has a profound impact on cellular metabolism, particularly by impairing mitochondrial respiration and promoting a shift towards glycolysis. This vulnerability to glycolysis inhibition, coupled with FCF-induced changes in various signaling pathways, suggests its potential as a therapeutic agent in endometrial cancer.

## Figures and Tables

**Figure 1 cancers-16-00976-f001:**
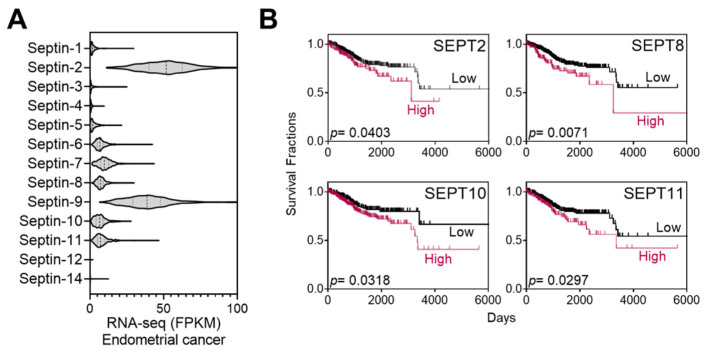
Expression of septins in endometrial cancer and Kaplan–Meier analysis. (**A**) TCGA database analyses of septin expression in endometrial cancer. (**B**) High-septin expressors in (**A**) (septin 2 and 6–11) were further analyzed by the Kaplan–Meier survival analyses. Among those, the association of SEPT6, 7, or 9 is statistically insignificant (*p* > 0.05) with patient mortality while the expression of SETP2, 8, 10, and 11 correlates with increased mortality.

**Figure 2 cancers-16-00976-f002:**
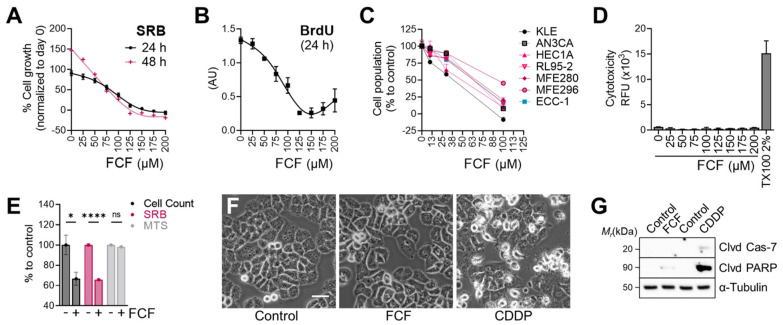
Effect of FCF on the proliferation of endometrial cancer cell lines. (**A**) MFE296 cells were treated with indicated concentrations (μM) of FCF for 24 or 48 h. After treatment, cells were fixed, and their population was measured by the sulforhodamine B (SRB) assay. Cell proliferation is described as a percent increase in cell population compared to that of fixed cells on day 0 (mean ± SEM, N = 3). (**B**) MFE296 cells were treated as in (**A**). Cell proliferation was monitored by DNA replication using the 5-bromo-2′-deoxyuridine (BrdU) assay (N = 3). (**C**) a panel of endometrial cancer cell lines was treated with FCF for 48 h. Cell population was measured by the SRB assay as described in (**A**). Calculated 50% inhibitory concentrations for each cell line as follows: KLE (38.8 µM), AN3CA (63.7 µM), HEC1A (48.8 µM), RL95-2 (63.2 µM), MFE280 (66.7 µM), MFE296 (92.5 µM), ECC-1 (63.4 µM) (**D**) cytotoxicity was measured by changes in cell membrane integrity as described in “Materials and Methods”. MFE296 cells were treated with indicated concentrations (μM) of FCF for 24 h. Triton X-100 (TX100, 2%) was included as a positive control for induced cytotoxicity. (**E**) MFE296 cells treated with FCF (100 μM, 24 h) were divided into three groups, of which cell counting, SRB, or MTS assay was performed to measure cell population. N = 6, ns: not significant, *: *p* = 0.0163, ****: *p* < 0.0001. (**F**,**G**) MFE296 cells were treated with FCF (100 μM) or cisplatin (CDDP), an apoptosis inducer, for 24 h, after which images were taken (at 20× objective; bar: 50 μm) to determine morphologic features of apoptosis (shrunk and rounded shapes) (**F**) or cells were lysed, separated by SDS-PAGE electrophoresis, and immunoblotted against apoptosis markers (Clvd Cas-7: cleaved caspase-7; Clvd PARP: cleaved PARP) and a loading control α-tubulin (**G**).

**Figure 3 cancers-16-00976-f003:**
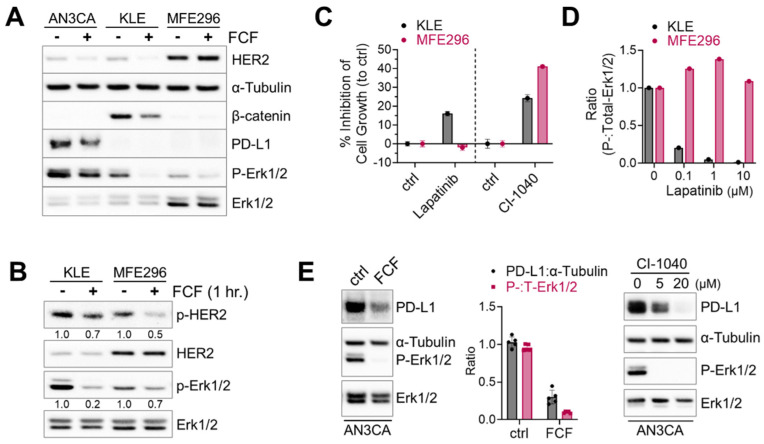
Inhibitory effect of FCF on Erk1/2 and HER2 phosphorylation. (**A**) Three different endometrial cancer cell lines, AN3CA, KLE, and MFE296 were treated with FCF (100 μM, 48 h). Protein expression was determined by immunoblotting using specific antibodies as indicated. (**B**) Immunoblotting of KLE and MFE296 cells treated with FCF (100 μM, 1 h) against Erk1/2, HER2, phospho-Erk1/2 (T202/Y204), and phospho-HER2 (Y1248). (**C**) KLE and MFE296 cells were treated with lapatinib (1 μM; N = 6) or CI-1040 (15 μM; N = 4) for 24 h. Cell population was measured by the SRB assay. (**D**) Immunoblotting of cells treated with lapatinib (μM, 2 h; N = 1). (**E**) Immunoblotting of AN3CA cells treated with FCF (100 μM) or CI-1040 (μM) for 24 h.

**Figure 4 cancers-16-00976-f004:**
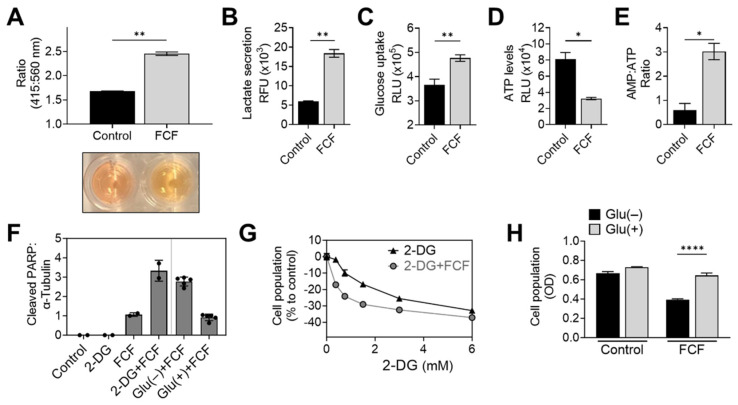
FCF alters cellular energy metabolism. (**A**) A pH change in cell culture media was accessed spectrophotometrically. KLE cells were treated with DMSO or FCF (100 μM) for 24 h, after which culture medium was collected and changes in phenol red absorbance was measured using a ratiometric analysis at 415 and 560 nm. 560 nm is more pronounced at basic pH while 415 nm is at acidic pH. (**B**) Analysis of lactate secretion into culture medium. KLE cells were treated with FCF (100 μM) overnight. The concentration of lactate in supernatants was measured by enzyme-based lactate oxidation and concomitant NAD^+^ reduction to NADH. (**C**) Effect of FCF on glucose uptake. MFE296 cells were treated with FCF for 5 h. Glucose uptake was measured by the addition of a glucose mimic 2-deoxyglucose (2-DG). 2-DG uptake was determined by a bioluminescence-based coupled enzymatic assay as described in “Materials and Methods”. (**D**,**E**) MFE296 cells were treated with DMSO or FCF (100 μM) for 18 h. Cell extracts were analyzed for ATP and AMP levels as described in “Materials and Methods”. ATP levels were normalized to the protein concentration of respective sample (**D**). (**F**) MFE296 cells were treated with 2-DG (12 mM) or cultured without glucose for 24 h in the presence of FCF (100 µM); Glu(−): glucose deprived, Glu(+): glucose (1 g/L) supplemented. Cell extracts from each condition were analyzed by immunoblotting. (**G**) As in (**F**), but treated with various concentrations of 2-DG (mM). The SRB assay was used to measure cell population. (**H**) As in (**F**). Cell population was determined by the SRB assay. (*: *p* < 0.05, **: *p* < 0.01, ****: *p* < 0.0001).

**Figure 5 cancers-16-00976-f005:**
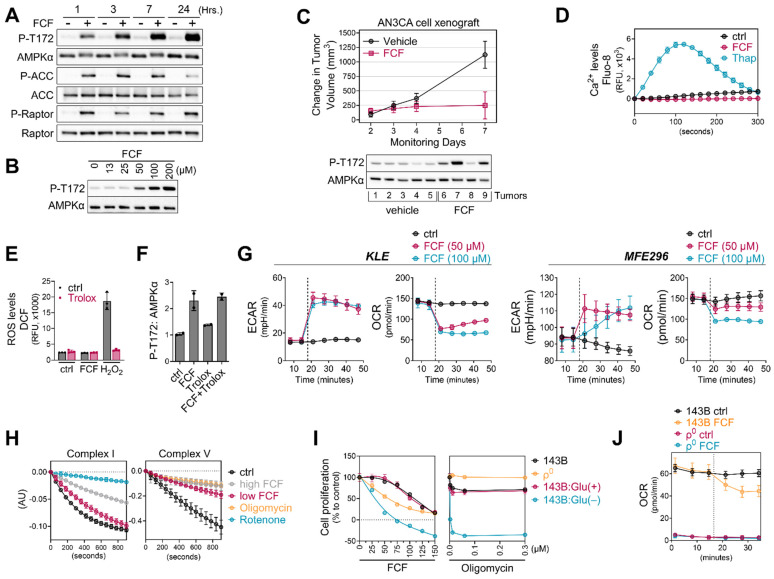
FCF impairs mitochondrial respiration. (**A**,**B**) MFE296 cells were treated with fixed concentration of FCF (100 μM) for indicated durations (**A**) or for 7 h with various concentrations (μM) of FCF (**B**). Immunoblotting was performed to measure total or phosphorylated AMPK (T172), ACC (S79), and Raptor (S792). (**C**): NSG mice implanted with AN3CA tumors were treated with vehicle (N = 5) or FCF (N = 5, 25 mg/kg, IP, M-F). The average change (+/− 1 SE) in tumor volume from the initiation of treatment estimated from a mixed model (top). After the treatments, AN3CA xenograft tumors were harvested and homogenized using a polytron in radioimmunoprecipitation assay buffer. The resulting mixture was then centrifuged, and the cleared protein extracts were separated by electrophoresis and subjected to immunoblotting using specific antibodies as indicated (bottom). (**D**) Drug-induced Ca^2+^ release in MFE296 cells was monitored by a fluorescent Ca^2+^ binding dye (Fluo-8) as described in “Materials and Methods”. Thapsigargin (Thap) that raises intracellular Ca^2+^ was included as a positive control. (**E**) Effect of FCF on oxidative stress. MFE296 cells were treated with FCF (100 μM) and hydrogen peroxide (H_2_O_2_, 100 μM) alone or in combination with an antioxidant Trolox (1 mM) for 2 h. The cellular levels of reactive oxygen species (ROS) were measured spectrophotometrically using a carboxy-H_2_DCFDA probe. (**F**) As in (**E**), immunoblotting used for the detection of AMPK phosphorylation (T172). (**G**) Oxygen consumption rate (OCR) and extracellular acidification (ECAR) measurement in KLE and MFE296 cells as described in “Materials and Methods”. Dot line: drug injection. (**H**) Effect of FCF on mitochondrial complex I and V. Isolated mitochondria were treated with vehicle, high (300 μM) and low (100 μM) concentrations of FCF, as well as established mitochondrial inhibitors: rotenone (2 μM) targeting complex I, and oligomycin (12.6 μM) targeting complex V. Complex I and V activities were measured as described in “Materials and Methods” monitoring NADH oxidation at 340 nm. (**I**) 143Bρ^0^ and 143B cells were treated with FCF or oligomycin at a range of concentrations for 24 h. Cell proliferation was determined by the SRB assay (compared to day 0). Glu(+): glucose (at 4.5 g/L) supplemented; Glu(−): glucose deprived. (**J**) OCR measured in 143B and 143Bρ^0^ cells in the presence of DMSO or FCF. Dot line: drug injection.

**Figure 6 cancers-16-00976-f006:**
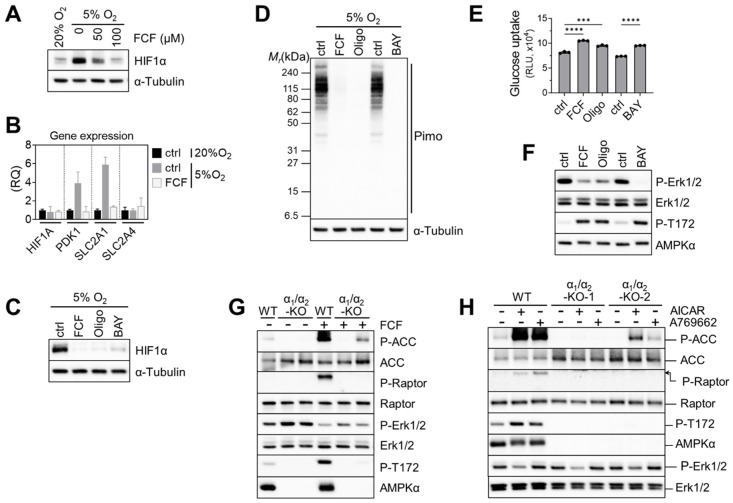
Impaired mitochondrial respiration by FCF causes HIF1α instability and glucose uptake in MFE296 cells. (**A**) Cells were cultured under 20% or 5% oxygen in the presence of FCF for 24 h. Cell extracts were collected and subject to immunoblotting with indicated antibodies. (**B**) Effect of FCF on hypoxia induced genes. Cells were treated as in (**A**), but for 6 h with FCF (100 μM), after which gene expression of HIF1A, PDK1, SLC2A1 (GLUT1), and SLC2A4 (GLUT4) were analyzed by quantitative RT-PCR. (RQ: relative quantification). (**C**) As in (**A**), but cells were treated with FCF or mitochondrial respiratory chain inhibitor for 6 h; oligomycin (1.5 μM, a complex V inhibitor) or BAY87-2243 (1 μM, a complex I inhibitor). (**D**) As in (**C**), but cells were co-treated with pimonidazole (20 μM) for 24 h. Pimonidazole is reductively activated during oxygen deprivation and generates protein adducts, which are recognized by a specific mouse monoclonal antibody. (**E**) Effect of mitochondrial inhibitors on glucose uptake. MFE296 cells were treated with FCF (100 μM), oligomycin (1.5 μM), or BAY87-2243 (1 μM) for 6 h. Glucose uptake was measured as described in Figure 4C. (***: *p* < 0.001, ****: *p* < 0.0001) (**F**) effect of mitochondrial inhibitors on Erk1/2 regulation. Cells were treated as in (**E**) and subjected to immunoblotting with indicated antibodies. (**G**,**H**) Wild-type MFE296 (WT) or double AMPK-α_1_/α_2_ knockout (α_1_/α_2_-KO) clones were incubated with FCF for 2 h (**G**) or with AMPK activators, AICAR (1 mM) or A769662 (200 μM), for 6 h (**H**). Total and phosphorylated proteins were analyzed by immunoblotting against AMPK (T172), ACC (S79), Raptor (S792), and Erk1/2 (T202/Y204).

**Figure 7 cancers-16-00976-f007:**
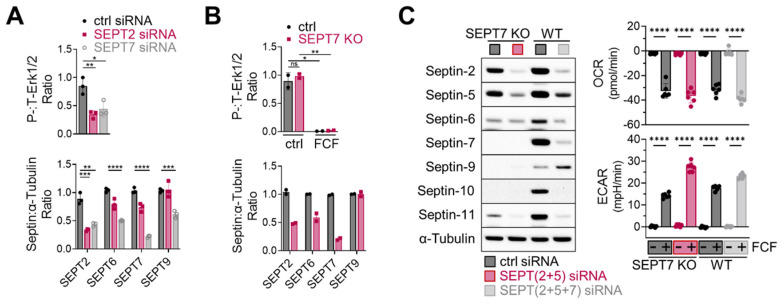
Effect of septin depletion in MFE296 cells. (**A**) Transient knockdown of septin-2 and -7 by siRNAs. (**B**) Septin-7 knockout using the CRISPR/Cas9 system. Cells were transfected with control or septin-7 double nickase plasmid. Following puromycin selection, polyclonal cells were treated with DMSO or FCF (100 μM, 6 h) and subject to immunoblotting with indicated antibodies. (**C**) Effect of FCF on OCR and ECAR in septin depleted cells. Wild-type MFE296 or single cell derived septin-7 knockout was transfected with siRNAs as indicated. After 48 h, cells were split into 96-well plates (for Seahorse analysis) or 6-well plates (for immunoblotting) and allowed to adhere overnight. Afterwards, expression of septins was analyzed by immunoblotting with specific antibodies to each septins (left), or OCR and ECAR were monitored following DMSO or FCF (100 µM) treatment. *y*-axis: changes after DMSO or FCF injection (right). (*: *p* < 0.05, **: *p* < 0.01, ***: *p* < 0.001, ****: *p* < 0.0001)

**Figure 8 cancers-16-00976-f008:**
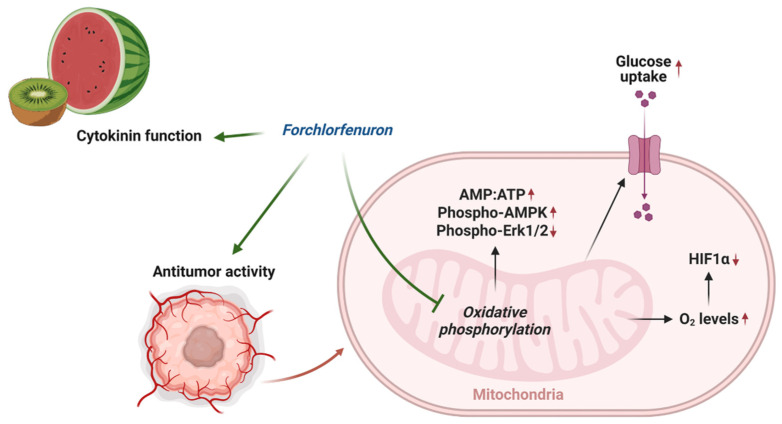
Overview of mechanism of action of FCF.

## Data Availability

The genomics data utilized in this study can be accessed via http://www.proteinatlas.org.

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
