# Peer review of "Forchlorfenuron-Induced Mitochondrial Respiration Inhibition and Metabolic Shifts in Endometrial Cancer"

_cancers, 2024, doi:10.3390/cancers16050976_

Round 1
Reviewer 1 Report
Comments and Suggestions for Authors
1. In the abstract, “the precise target of FCF has yet to be fully determined” should be changed to “….. has not yet to be fully determined”.
2. “This study reveals a novel target of FCF and elucidate its downstream”, “elucidate” should be “elucidates”.
3. In your figures, whenever you use small molecule to treat cells, there should be a unit for this small molecule, such as μM, nM, etc.
4. I suggest you calculate and inform the IC50 for FCF on different cells you used in this article.
5. From your figure 3, it seems FCF is not that much putative for these cancer cells since you treated 48hours with 100μM, this is a long time at high dose.
6. You mentioned that you used seahorse analyzer to determine the OCR and ECAR after FCF treatment of two cancer cells, please put the figures. Figure 5G is not the classical one and it is not clear that the OCR level is basel or after oligomycin A, FCCP, ROT/AA treatment. Please re-do the experiment and change to correct figures. Again, any time when you use FCF in your figures, please mark the dose.
7. You’ve done lots experiments to indicate that FCF targets mitochondria, but the most important point is to elucidate the interacting protein of FCF, which is the real target. You need to use some methods such as pull-down+LC/MS or Proximity labeling to identify the exact protein interacted with FCF, and if it is a mitochondria protein, or it can regulate the OXPHOS pathway, then you add all these related experiments, you can get the conclusion.
8. The language editing should be done for this article.
9. The FCF treatment on MFE296 and KLE indicated that it can reduce OCR at a high dose instead of a low dose, which indicates that its real target may not be in mitochondria. In addition, cancer cells are very heterogeneous, some are very sensitive to OXPHOS-inhibition and some others are resistant, and more are on the middle. You need to use more cancer cell line to indicate that FCF inhibit OCR in cancer cells.
10. You really did lots of experiments in this article, but the most important scientific question is still pending to be answered. The most important experiment should be done, the exact target of FCF should be identified and verified.
Comments on the Quality of English LanguageShould be improved
Author Response
Reviewer #1
- In the abstract, “the precise target of FCF has yet to be fully determined” should be changed to “….. has not yet to be fully determined”. Response: The sentence has been corrected.
- “This study reveals a novel target of FCF and elucidate its downstream”, “elucidate” should be “elucidates”. Response: The word has been corrected.
- In your figures, whenever you use small molecule to treat cells, there should be a unit for this small molecule, such as μM, nM, etc. Response: We have included the appropriate units in the figures.
- I suggest you calculate and inform the IC50 for FCF on different cells you used in this article. Response: We have included the calculated 50% inhibitory concentrations for each cell line in the figure legend corresponding to Figure 2.
- From your figure 3, it seems FCF is not that much putative for these cancer cells since you treated 48hours with 100μM, this is a long time at high dose. Response: I agree. The FCF treatment duration of 24-48 hours at a concentration of 100 μM, as employed in our current study, may appear prolonged and administered at a high dose. However, it is important to note that these concentrations fall well within the established and widely accepted range for the treatment of mammalian cells. FCF is typically utilized at concentrations ranging from 50-200 µM for mammalian cell studies, and in some cases, even higher concentrations such as 1-2 mM are employed for certain species (as documented in PMID: 27473917 and other references within). Despite this limitation, FCF still remains a primary choice for studying septins and treating mammalian cells, as there are no other small molecule inhibitors readily and commercially available at this point. The urgent need for more efficacious and selective septin inhibitors is evident.
- You mentioned that you used seahorse analyzer to determine the OCR and ECAR after FCF treatment of two cancer cells, please put the figures. Figure 5G is not the classical one and it is not clear that the OCR level is basel or after oligomycin A, FCCP, ROT/AA treatment. Please re-do the experiment and change to correct figures. Again, any time when you use FCF in your figures, please mark the dose. Response: As per your suggestion, Figure 5G has been updated. The new figures show OCR and ECAR monitored over time, along with basal level information (Figure 5G and supplemental Figure S5). Additionally, we have added information on drug concentrations. The text has been revised to address these changes.
- You’ve done lots experiments to indicate that FCF targets mitochondria, but the most important point is to elucidate the interacting protein of FCF, which is the real target. You need to use some methods such as pull-down+LC/MS or Proximity labeling to identify the exact protein interacted with FCF, and if it is a mitochondria protein, or it can regulate the OXPHOS pathway, then you add all these related experiments, you can get the conclusion. Response: FCF is frequently used as a pharmacological tool to study septins; however, the precise cellular and molecular mechanisms responsible for FCF's effects have remained elusive. In our study, we aim to provide a rapid communication that contributes to the understanding of FCF's mechanism of action within the research community. Through our current investigation, we have established that FCF effectively inhibits mitochondrial respiration, which is previously unknown. We believe that our findings are pivotal in elucidating previously unexplained phenomena, such as the degradation of HIF-1α (PMID: 23977378) and the enhanced glucose uptake (PMID: 22809625) observed in earlier studies. To support our findings, we conducted enzymatic assays targeting mitochondrial complexes on isolated mitochondria. Additionally, we employed the Seahorse analyzer to measure cellular mitochondrial respiration and explored other bioenergetic properties influenced by FCF, such as glucose uptake, lactate secretion, extracellular acidification, changes in AMP:ATP ratio and the activation of AMPK. Our results consistently indicate that FCF exerts its effects by inhibiting mitochondrial respiration. We acknowledge the importance of identifying specific interacting proteins of FCF within the mitochondria. Techniques such as pull-down assays coupled with LC/MS or proximity labeling should offer more detailed information about the specific targets of FCF. While we appreciate the importance of the proposed experiments, these experiments may extend beyond the scope of our research focus. Nevertheless, our results provide valuable insights for researchers studying FCF and create a foundation for further clarifying specific targets of FCF.
- The language editing should be done for this article. Response: Thank you for your suggestion. We have revised the language in the article to enhance its clarity.
- The FCF treatment on MFE296 and KLE indicated that it can reduce OCR at a high dose instead of a low dose, which indicates that its real target may not be in mitochondria. In addition, cancer cells are very heterogeneous, some are very sensitive to OXPHOS-inhibition and some others are resistant, and more are on the middle. You need to use more cancer cell line to indicate that FCF inhibit OCR in cancer cells. Response: We have conducted new experiments using three other cancer cell lines, and the effects of FCF on OCR and ECAR remain consistent with our findings in MFE296 and KLE. We have incorporated the new data into supplemental Figure S5 and revised the text accordingly.
- You really did lots of experiments in this article, but the most important scientific question is still pending to be answered. The most important experiment should be done, the exact target of FCF should be identified and verified. Response: Please refer to our response addressing Question 7 above.
Reviewer 2 Report
Comments and Suggestions for Authors
The manuscript titled "Forchlorfenuron-Induced Mitochondrial Respiration Inhibition and Metabolic Shifts in Endometrial Cancer" authored by Kyukwang Kim et al. reveals a novel target of FCF and elucidates its downstream signaling events, suggesting FCF's potential as a therapeutic option in endometrial cancer. While well-written, the manuscript requires attention to several issues before further consideration.
1. The assertion that FCF alters energy metabolism in endometrial cancer cells lacks sufficient support from current results. The authors examined lactate, glucose, and ATP levels using only one inconsistent cell line under each condition. Additional results from another cell line are necessary. Figure 4F needs clearer explanation, and WBs data is requested. Furthermore, for results from Figure 4G, the cell population from 2-DG treatment should be included in the figure.
2. Validation results for the 143Bρ0 cells are requested.
3. Figure 3 suggests that MFE296 cells behave differently compared to AN3CA and KLE. The selection of MFE296 cells in figures 6 and 7 needs justification. Are these results specific to this cell line? Clarification is required.
Minor Comments:
1. The figures should be ordered according to the sequence mentioned in the manuscript.
2. In Figure 1, add the size of the dataset to both figures A and B.
3. Figure 2A shows % cell growth up to 150%, which requires explanation. Figure 2F fails to reflect apoptosis adequately. The statement "Our data shows that FCF is not a potent inducer of apoptosis (Fig. 2F and 2G), nor of cytotoxicity (Fig. 2D) in MFE296 cells" needs clarification. Regarding how cell viability is affected, FCF should have cytotoxicity.
4. In Figure 3, the statement "Concurrently, FCF led to an increase in tubulin acetylation, potentially linked to septin disruption" needs supporting evidence. Otherwise, this data should remove to supplementary.
5. In Figures 5 and 8, please check Phosphate-AMPK on WBs. D, hard to tell the two pink color. Please check H, control should have minimal reduction?
6. Consider including strong results from at least two cell lines and moving some results to supplementary figures for improved clarity.
Comments on the Quality of English Language
The Quality of English Language is fine.
Author Response
Reviewer #2
The manuscript titled "Forchlorfenuron-Induced Mitochondrial Respiration Inhibition and Metabolic Shifts in Endometrial Cancer" authored by Kyukwang Kim et al. reveals a novel target of FCF and elucidates its downstream signaling events, suggesting FCF's potential as a therapeutic option in endometrial cancer. While well-written, the manuscript requires attention to several issues before further consideration.
- The assertion that FCF alters energy metabolism in endometrial cancer cells lacks sufficient support from current results. The authors examined lactate, glucose, and ATP levels using only one inconsistent cell line under each condition. Additional results from another cell line are necessary. Figure 4F needs clearer explanation, and WBs data is requested. Furthermore, for results from Figure 4G, the cell population from 2-DG treatment should be included in the figure. Response: As per your suggestion, we have added new data (Figure S2, S3, and S4) from another cell line and incorporated them into the manuscript. The text and figures have been updated to reflect these changes. Figure 4F displays the densitometric analysis obtained from the western blot assay, as suggested. The original images of the western blot are provided in the supplemental data for reference. We have revised Figure 4G for better clarity as per your suggestion.
- Validation results for the 143Bρ0 cells are requested. Response: The manuscript has been revised to incorporate the following updates: The RT-PCR results, as depicted in Supplemental Figure S6, show a significant loss of mitochondrial DNA in the 143Bρ0 cells. Additionally, the Seahorse analysis presented in Figure 5J demonstrates impaired mitochondrial respiration in these 143Bρ0 cells.
- Figure 3 suggests that MFE296 cells behave differently compared to AN3CA and KLE. The selection of MFE296 cells in figures 6 and 7 needs justification. Are these results specific to this cell line? Clarification is required. Response: Thank you for your suggestion. We acknowledge that the previous paragraph lacked clarity. Therefore, we have revised the paragraph corresponding to Figure 3 to provide a clearer explanation and justification for selecting MFE296 cells for subsequent studies (Figures 6 and 7). We described that FCF, along with other mitochondrial respiratory inhibitors, influences Erk1/2 signaling in MFE296 cells (Figure 6). The impact of FCF on mitochondrial respiration was observed across multiple cell lines, including MFE296 and KLE (Figure 5G) in the main text, with an additional three cell lines incorporated into the supplemental material (Figure S5).
Minor Comments:
- The figures should be ordered according to the sequence mentioned in the manuscript. Response: The figures, especially Figure 3, have been rearranged in accordance with the mentioned sequence, and the text and figure legend have been revised accordingly.
- In Figure 1, add the size of the dataset to both figures A and B. Response: The size of the dataset has been incorporated into the Materials and Methods section.
- Figure 2A shows % cell growth up to 150%, which requires explanation. Figure 2F fails to reflect apoptosis adequately. The statement "Our data shows that FCF is not a potent inducer of apoptosis (Fig. 2F and 2G), nor of cytotoxicity (Fig. 2D) in MFE296 cells" needs clarification. Regarding how cell viability is affected, FCF should have cytotoxicity. Response: In Figure 2A, the data has been normalized to the cell population on day 0, wherein 150% cell growth signifies a 150% increase relative to the baseline on day 0. FCF does not induce cytotoxicity or apoptosis in MFE296 cells (Figure 2D, 2F, and 2G). However, it does exhibit an inhibitory effect on cell proliferation (Figure 2A and 2B).
- In Figure 3, the statement "Concurrently, FCF led to an increase in tubulin acetylation, potentially linked to septin disruption" needs supporting evidence. Otherwise, this data should remove to supplementary. Response: We have moved this figure to the supplementary section (Figure S1), and the text and Figure 3 have been revised accordingly.
- In Figures 5 and 8, please check Phosphate-AMPK on WBs. D, hard to tell the two pink color. Please check H, control should have minimal reduction? Response: Phospho-AMPK is labeled as Phospho-T172 to indicate the specific activity site of AMPK. In Figure 5D, we modified the color scheme. In Figure 5H, the rate of NADH oxidation is measured by a decrease at 340 nm and is proportional to the activity of mitochondrial complexes.
- Consider including strong results from at least two cell lines and moving some results to supplementary figures for improved clarity. Response: We have added new data (Figure S2, S3, and S4) from another cell line. The manuscript has been revised accordingly.
Round 2
Reviewer 1 Report
Comments and Suggestions for Authors
Figure 5G, I requested to redo the experiment of seahorse and mark the OCR and ECAR changed under treatment of FCF, oligomycinA, FCCP, ROT/AA. The author did not do this.
Author Response
Comment: Figure 5G, I requested to redo the experiment of seahorse and mark the OCR and ECAR changed under treatment of FCF, oligomycinA, FCCP, ROT/AA. The author did not do this.
Response:
-
I sincerely apologize for the oversight in my previous response. The primary focus of our study is to promptly share our novel findings that FCF inhibits mitochondrial respiration, a previously unpublished observation.
-
Our experiment did not include the mitochondrial stress test involving sequential treatment with Oligomycin, FCCP, Rotenone/Antimycin. We acknowledge the importance of this test as it provides a more comprehensive understanding of FCF's impact on mitochondria, covering basal and maximal respiration, ATP production, and other parameters. However, the measurement of OCR and ECAR often occurs independently of the mitochondrial stress test (PMIDs: 24843020, 24280423, and 26534958).
- In addition to our investigation into FCF's impact on OCR using the Seahorse analyzer, we also explored other bioenergetic properties affected by FCF, such as glucose uptake, lactate secretion, extracellular acidification, changes in the AMP:ATP ratio, and the activation of AMPK. Consistently, our results indicate that FCF exerts its effects by inhibiting mitochondrial respiration.
-
While we acknowledge the significance of the mitochondrial stress test, we propose that future Seahorse analyzer studies incorporate this test alongside FCF. Additionally, studies could involve different protocol settings using various mitochondrial inhibitors and substrates that target specific respiratory complexes (PMIDs: 27506290 and 36358872).